

# Profiling of primary and phytonutrients in edible mahlab cherry (*Prunus mahaleb* L.) seeds in the context of its different cultivars and roasting as analyzed using molecular networking and chemometric tools

Mayy M. Mostafa and Mohamed A. Farag

Pharmacognosy, Cairo University, Cairo, Egypt

## ABSTRACT

*Prunus mahaleb* L. (mahlab cherry) is a deciduous plant that is native to the Mediterranean region and central Europe with a myriad of medicinal, culinary and cosmetic uses. The present study explored different cultivars of mahlab (white from Egypt and Greece, red from Egypt and post roasting). UPLC-MS led to the detection of 110 primary and secondary metabolites belonging to different classes including phenylpropanoids (hydroxy cinnamates, coumaroyl derivatives), organic acids, coumarins, cyanogenic glycosides, flavonoids, nitrogenous compounds, amino acids and fatty acids, of which 39 are first time to be detected in *Prunus mahaleb* L. A holistic assessment of metabolites was performed for further analysis of dataset using principal component analysis (PCA) among mahlab cultivars to assess variance within seeds. The results revealed that phenolic acids (coumaric acid-*O*-hexoside, ferulic acid-*O*-hexoside, ferulic acid-*O*-hexoside dimer, dihydrocoumaroyl-*O*-hexoside dimer and ferulic acid), coumarins (coumarin and herniarin) and amino acids (pyroglutamic acid) were abundant in white mahlab cultivars (cvs.) from different locations. In contrast, red mahlab and its roasted seeds were more rich in organic acids (citric and malic acids), amygdalin derivative and sphingolipids. Orthogonal projections to latent structures discriminant analysis (OPLS-DA) revealed for markers in red mahlab and in response to roasting, where red mahlab was rich in nitrogenous compounds viz. nonamide, deoxy fructosyl leucine, glutaryl carnitine and isoleucine, while roasted product (REM) was found to be enriched in choline.

# INTRODUCTION

*Prunus mahaleb* L. (*P. mahaleb* L.) is a deciduous tree that belongs to family Rosaceae. It is indigenous to the Mediterranean region, Central Europe and West Asia. *P. mahaleb* L. is also known as mahlab, mahaleb cherry tree, the rock cherry as well as St. Lucie cherry. The tree is found wild in nature or cultivated as ornamental plants (*Özçelik et al., 2012*).

Corresponding author
Mohamed A. Farag,
mohamed.farag@pharma.cu.edu.eg

Being valuable, the mahaleb tree is widely cultivated in different countries. In Saudi Arabia and Sudan, the crushed seeds and kernels of mahlab are used as cosmetics in wedding preparations or as a nourishing oil for hair (*Sbihi, Nehdi & Al-Resayes, 2014*).

The plant is used in the production of fragrances, lotions, liquors and vinegar in food industry. Moreover, the kernels impart characteristic bitter taste comparable to bitter almonds when used as flavoring agent in cakes, biscuits, bagels, candies and cookies (*Gerardi et al., 2015*). Small amounts of powdered seeds are used in several countries such as Turkey, Armenia and Greece to enhance the taste of sweet foods (*Gerardi et al., 2015*).

Traditionally, fruits and seeds are used in diabetes and gastrointestinal problems (*Özçelik et al., 2012*). When considering its chemical composition, mahlab fruits are vital source of polyphenols such as anthocyanins explaining its use as raw material for functional food (*Blando et al., 2016*). *P. mahaleb* L. was reported for the presence of coumarin and herniarin (7-methoxy coumarin) to account for the antioxidant activity of the plant in addition to pleasant aroma (*Farag et al., 2021*). Moreover, the kernels and whole fruits of *P. mahaleb* L. were also reported to encompass high anthocyanin content particularly cyanidin 3,5-*O*-diglucoside, cyanidin-3-*O*-sambubioside, cyanidin-3-*O*-xylosyl-rutinoside and cyanidin-3-*O*-rutinoside (*Blando et al., 2016*). With regard to nutrients composition, mahlab kernels are rich source of proteins (30.98%) and fatty oil (40.4%) (*Mariod et al., 2009*).

The seeds is enriched in oil that is rich in polyunsaturated fatty acid containing mainly $\alpha$-eleostearic, linoleic and oleic acids (*Farag et al., 2021*). Besides, mahlab was reported to exert anti-inflammatory, anti-oxidant, antimicrobial and antifungal activities (*Farag et al., 2021*). In regard to the differences among mahlab cultivars, it was reported that white mahlab (*Prunus mahaleb*) seeds are more rich in oil and protein than black mahlab (*Monechma ciliatum*) seeds (*Mariod et al., 2009*). Moreover, another study on white mahlab from Egypt revealed that it is distinguished by its highest fatty acid level (oleic acid, $\alpha$-linoleic acid) compared to red mahlab and white mahlab obtained from different origins (Sudan and Greece). In addition, white mahlab possessed the highest omega 3 to 6 ratio with high levels of omega 9 oleic acid and lower sugar composition compared to the above mentioned cultivars (*Farag et al., 2021*).

Our aim is to assess the compositional diversity in the chemical profile and nutrients of *Prunus mahaleb* L. seeds in accordance to its cultivar type and further in response to roasting. This is the first study of LCMS metabolomics and molecular networking to profile red and white mahlab (*Prunus mahaleb*) seeds from two different origins in the Mediterranean region where the tree grows, and further in response to roasting. Advanced chemical profiling such as liquid chromatography (LC-MS) or gas chromatography coupled to mass spectrometry (GC-MS) has recently shown remarkable contribution towards food and functional food analysis in emerging food metabolomics owing for their sensitivity and structural elucidation power (*El-Akad, El-Din & Farag, 2023*). We have previously reported on the use of GC-MS for aroma profiling in *P. mahaleb* L. in context to its cultivar type and roasting (*Farag et al., 2021*). The work herein is aimed at investigating heterogeneity in different *P. mahaleb* L. cultivars (cvs.) and post-roasting using UPLC-MS-based metabolomics with regards to secondary metabolites' composition

more likely to contribute to seeds' health benefits. Compared to LC-MS, UPLC-MS (ultra performance liquid chromatography coupled to mass spectrometry) provides improved peak separation with higher sensitivity level aiding to resolve seeds metabolome and employed in this study for the first time for profiling of *P. mahaleb* L. seed.

## MATERIAL AND METHODS

### Plant material

*Prunus mahaleb* L. seeds were obtained from Egypt, harvested in late summer during the year of 2019. Commercial samples from Greece were also obtained in 2019. The samples were coded as follows: white seeds obtained from Athens, Greece (EGM), white seeds obtained from Cairo, Egypt (WEM), red seeds obtained from Cairo, Egypt (REM) and roasted red seeds (RREM).

The powdered seeds were roasted by wrapping 5 g in aluminum foil and then heating it on a heating mantle at 180 °C for 15-20 min. The extraction procedure and sample preparation are discussed in details in supplementary file. A voucher specimen was deposited at the herbarium of the faculty of pharmacy, Cairo university, Cairo, Egypt. Each specimen was analyzed in triplicates to assess for biological replicates among seeds.

### UPLC-MS based multivariate data analysis and molecular networking

Full scan /automated MSMS in data dependent mode was acquired in both negative and positive modes for all files (100- 1400 *m/z*) using a UPLC system coupled online with a 6540 UHD Accurate-Mass Q-TOF (Agilent Technologies) following exact conditions described in *Saied & Farag (2023)* and *Sallam et al. (2022)*. Exact details are described in the Supplemental File. Molecular networks were generated for negative and positive ionization files applying Global Natural Products Social Molecular networking (GNPS, http://gnps.ucsd.edu/ProteoSAFe/status/gnps-splash.jsp) accessed date (14 April 2022). The parameters are mentioned in the Supplemental File.

## RESULTS AND DISCUSSION

The goal of our study was to examine metabolome diversity in white and red mahlab cvs. obtained from Egypt and Greece, and further in response to roasting. UPLC-ESI-MS analysis was performed in negative and positive ionization modes to provide a comprehensive overview of the metabolome in examined *P. mahaleb* L. cvs. (EGM, WEM, REM and RREM) resulting in the detection of 110 metabolites. Different classes of metabolites were annotated including phenylpropanoids (hydroxyl cinnamates, coumaroyl derivatives), organic acids, coumarins, cyanogenic glycosides, flavonoids, nitrogenous compounds/amino acids and fatty acids Table S1, Fig. 1 and Fig. S1). Metabolites were eluted based on polarity in descending order in negative (51 metabolites) and positive ionization modes (61 metabolites) providing better coverage of seeds' metabolome. Results revealed that negative ionization provided good representative for hydroxycinnamates, while nitrogenous compounds were better detected in positive ionization mode.

Global natural product social molecular networking (GNPS) was performed to visualize mahlab cvs. seed metabolome as detected using UPLC-ESI-MS. GNPS is a system that
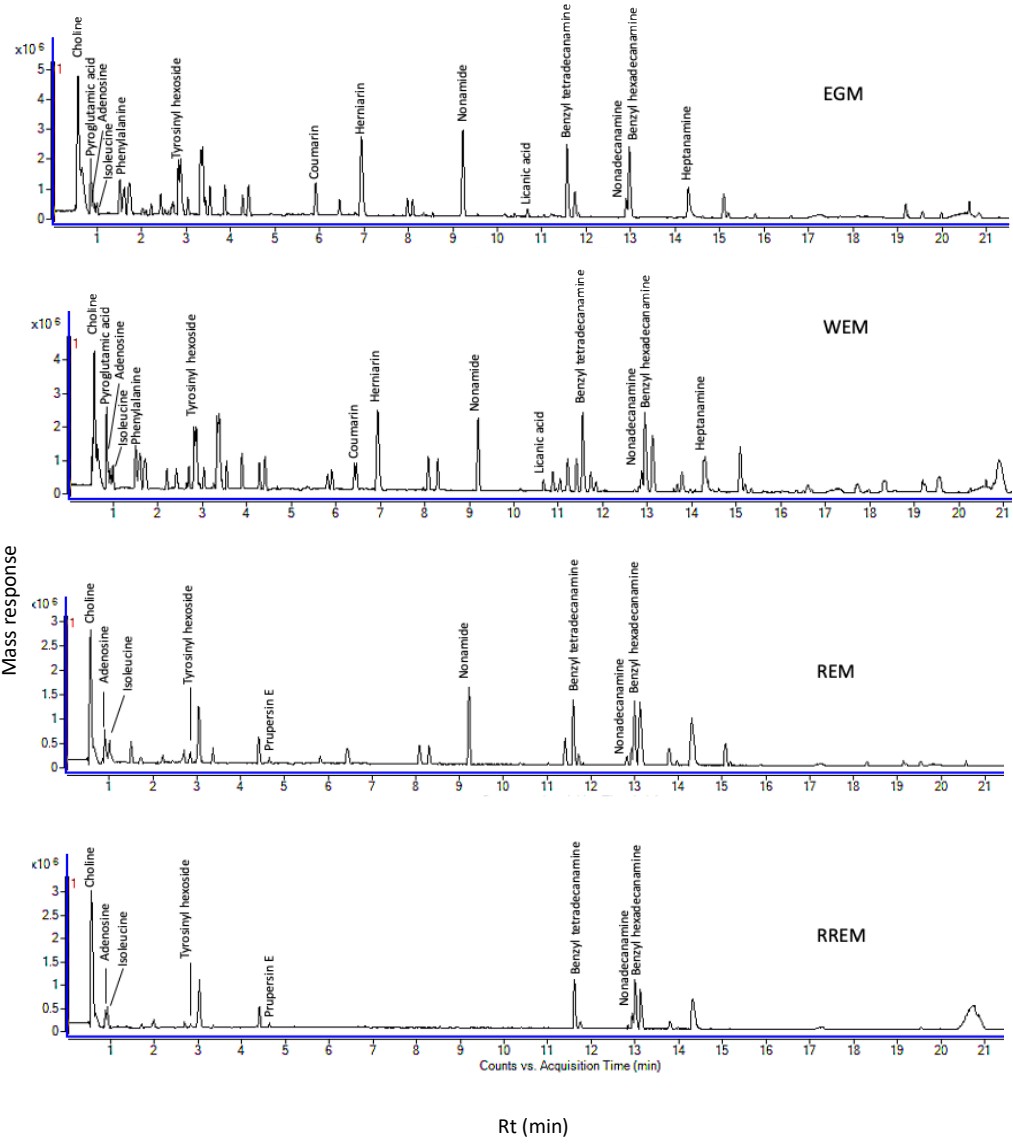

**Figure 1** Ultra-performance liquid chromatography-mass spectrometry (UPLC-MS) base peak chromatograph of *Prunus mahaleb* L. seeds methanol extract in positive ion mode.

calculates the scores between all the fragment ions (MS/MS) inside a dataset as an early step to analyze set of data in comparison with all public databases and to aid in peaks identification. The molecular networks generated by GNPS are considered as a visual inspection of a group of spectra of structurally related molecules aiding in metabolite annotation (*Shakour et al., 2023*). Moreover, the edges correspond to the alignment between spectrums, and connections between two nodes contributes to the formation of clusters of similar molecules to distinguish between the distinct families included in the network (*El-Hawary et al., 2022*). The nodes of the network represent compounds' parent

ions, whereas node colors represent the cultivars attributes provided from the metadata file.

Two molecular networks (MN) were extracted from negative and positive ionization modes tandem MS data. Nodes were represented by a pie chart to show the metabolites' abundance in the different cvs. The samples were coded with different colors, where green represents EGM, yellow represents WEM, purple represents REM and red represents RREM. Identified compounds belonging to phenylpropanoids, cyanogenic glycosides, amino acids, fatty acid ethanolamides, arylalkylamines and flavonoids; some of which were first time to be detected in mahlab cvs. and to add to its functional and health attributes. The negative MN encompassed 80 nodes distributed within 11 clusters. The negative MN was able to reveal hydroxycinnamates (cluster A) and cyanogenic glycosides (cluster B) considering their better detection in that mode.

In contrast, the positive molecular network (Fig. S2) was composed of 251 nodes grouped in 34 clusters. The significant sets of interest were cluster A (amino acids) and cluster B (coumarins). Interestingly, the positive MN was able to show classes for the first time in genus *Prunus* exemplified by cluster C (fatty acid ethanolamides) and cluster D (arylalkylamines). In addition, flavonoids were grouped in cluster E based on characteristic sugar and RDA losses in that class.

Amino acids, arranged in cluster A (positive MN, Fig. S2) were all detected in white mahlab cvs. (EGM and WEM), red mahlab (REM) except for arginine, (RREM) except for arginine and tyrosine. However, amino acids detected in roasted red mahlab were at comparable lower levels than unroasted red mahlab. Coumarins in cluster B were detected in all mahlab cvs. Fatty acid ethanolamides (cluster C) showed predominance of green color inferring for their abundance in white mahlab from Greece (EGM). Arylalkyl amines grouped in cluster D were detected at comparable levels in all cvs. Positive MN revealed flavonoids in cluster E, detected only in red mahlab and to likely contribute for seeds color. Moreover, representative structures and fragmentation patterns of major identified metabolites discussed in the manuscript are shown in Figs. S3–S11.

## Identification of metabolites
### Identification of hydroxycinnamates
Hydroxycinnamates are plant phenolic acids that are derived from cinnamic acids (i.e., caffeic, coumaric and ferulic acids) (*Kondrashev et al., 2020*) known to exert antibacterial activity against pathogens (*Esherichiacoli*, *Staphylococcus aureus* and *Bacillus cereus*), antioxidant activity, anticancer and antigenotoxic (*Kondrashev et al., 2020*). Hydroxycinnamates were the most common secondary metabolite class detected in mahlab cvs. in negative ionization mode as represented by 10 cinnamoyl derivatives (peaks 28, 39, 48, 51, 53, 55, 56, 58, 65 and 66), whereas only two phenolic acids (38 and 59) were detected in positive mode considering the improved ionization of phenolic acids in negative mode owing to their electronegative character.

Peak 38 with $[M+H]^+$ at *m/z* 165.0548 was identified as *p*-coumaric acid and only detected in EGM and WEM (*El-sayed et al., 2021*). Peaks 28 and 39 appeared with $[M-H]^-$ at *m/z* 487.1454 and 325.0785, respectively. The mass spectra of these peaks showed typical

fragmentation of *p*-coumaric acid with fragment ions at *m/z* 163 and 119. Moreover, fragment ion detected at *m/z* 325 confirmed the neutral loss of 162 amu explaining the loss of hexose unit for *p*-coumaric acid-*O*-dihexoside (peak 28) and *p*-coumaric acid-*O*-hexoside (peak 39) (*Blando et al., 2016*). *p*-Coumaric acid-*O*-dihexoside was only detected in EGM and WEM, while *p*-coumaric acid-*O*-hexoside was detected in all mahlab cvs. Peak 51 showed [M-H]$^-$ at *m/z* 327.1082 with subsequent loss of hexose moiety yielding fragment ion at *m/z* 165 [M-H-162]$^-$ and annotated as dihydrocoumaroyl-*O*-hexoside. Peak 53 exhibited [M-H]$^-$ at *m/z* 655.2164 followed by the loss of 328 amu displaying a fragment ion at *m/z* 327 and followed by fragmentation pattern similar to peak 51 suggesting for a dimer of dihydrocoumaroyl-*O*-hexoside in peak 53 (*Spínola et al., 2016*). Both compounds were only detected in EGM and WEM, alongside which dihydrocoumaroyl-*O*-hexoside dimer was the first time to be detected in *Prunus mahaleb* L. It should be noted that *p*-coumaric acid (*Orlando et al., 2021*), *p*-coumaric acid-*O*-hexoside, *p*-coumaric acid-*O*-dihexoside and dihydrocoumaroyl-*O*-hexoside (*Ieri, Pinelli & Romani, 2012*) were previously identified in fruits of *P. mahaleb*.

Peak 59 was identified as ferulic acid (Fig. S3) with [M+H]$^+$ at *m/z* 195.0652 with fragment ion at *m/z* 177 [M+H-H2O]$^+$ detected in most seed accessions viz. EGM, WEM, REM and RREM (*El-sayed et al., 2021*). Peaks 48 and 58 showed typical fragmentation pattern of ferulic acid, with peak 48 likewise identified as ferulic acid-*O*-hexoside dimer (Fig. S3) with [M-H]$^-$ at *m/z* 711.2147 that showed loss of fragment ion at *m/z* 356 and *m/z* 193.This compound was abundant in EGM and WEM. Likewise, peak 58 exhibited [M-H]$^-$ at *m/z* 355.1033 with the neutral loss of hexose (162 amu) yielding fragment ion at *m/z* 193 corresponding to ferulic acid and assigned as ferulic acid-*O*-hexoside (Fig. S3) that was present in all mahlab seed cvs. (*Baky et al., 2021*). Furthermore, peak 55 showed [M-H]$^-$ at *m/z* 357.1186 was assigned as hexosyl methoxy phenylpropanoic acid based on fragment ions at *m/z* 195 [(M-H)-162]$^-$ and 151 [(M-H)-(162+44)]$^-$ (*Aquino et al., 2002*). This compound was only detected in EGM and WEM. It was previously isolated from *P. mahaleb* L. seeds and to account for the bronchodilator activity of the plant (*Abdel-Kader et al., 2022*). Moreover, peak 56 and 66 were only assigned in EGM and WEM. Peak 56 was identified as dihydroferulic acid-*O*-hexoside dimer with [M-H]$^-$ at *m/z* 715.2462 and further fragment ions at *m/z* 327 and 195. Peak 66 with [M-H]$^-$ at *m/z* 195.006 was identified as dihydroferulic acid with main fragment ions at *m/z* 136 and 121 due to successive loss of decarboxylation and demethylation (*Yuan et al., 2021*). How these phenolic dimers are produced in mahlab has yet to be identified likely to be mediated *via* peroxidase catalyzed reaction as in legumes (*Farag et al., 2008*). Ferulic acid-*O*-hexoside dimer was the most prominent phenolic acid in white mahlab of both origin, found less abundant in red mahlab and to completely disappear in roasted red mahlab as a result of the roasting process. It's noteworthy to mention that white mahlab cvs. appeared to be generally more rich in hydroxycinnamic acids than red mahlab and expectedly its roasted seed. Other phenolics reported first time in *P. mahaleb* seed included ferulic acid-*O*-hexoside and its dimer, dihydroferulic acid-*O*-hexoside dimer and dihydroferulic acid and suggestive for the activation of ferulates biosynthesis in mahlab.

## Identification of organic acids

Organic acids and their salts improve the digestive process and the implementation of proteins in animals (*Farag et al., 2021*), in addition to acting as preservative and to influence taste in seeds. In the present study, organic acids were detected in seeds of all mahlab cvs., especially in negative mode considering the better ionization of these carboxyl containing compounds. Gluconic acid was detected in peak 5 with [M-H]⁻ at $m/z$ 195.051 in EGM and REM, whereas malic acid appeared in peak 8 with [M-H]⁻ at $m/z$ 133.0145 detected in WEM, REM and RREM, though found mostly abundant in red mahlab and its roasted product. Malic acid, an usual organic acid in unripe fruits, was reported previously in seeds of *P. mahaleb* (*Farag et al., 2021*). Peak 12 was identified as citric acid common in all mahlab seeds with [M-H]⁻ at $m/z$ 191.0197 (*Baky et al., 2021*) mostly abundant in REM and likely to contribute for seeds' flavor and shelf life (*Saha et al., 2013*). Other organic acid detected only in red mahlab included lactic acid in peaks 13 [M-H]⁻ at $m/z$ 89.0244. Peak 18 [M-H]⁻ at $m/z$ 117.0194 was assigned as succinic acid and detected in REM and RREM. Citric and succinic acid were reported previously in *P. mahaleb* (*Gerardi et al., 2015*), however, gluconic and lactic acids are first time to be detected in mahlab cvs. Phytic acid, an anti-nutrient which has a strong chelating property that hinders the bioavailability of proteins and minerals, was not detected in any of the seeds, which indicates their potential nutrient quality (*Farag et al., 2021*).

## Identification of coumarins

Coumarins are natural phenolics to encompass benzo-pyranone nucleus, and provide aromatic and fragrant characteristics compared to vanilla (*Lončar et al., 2020*). Also, coumarins exhibit a wide range of pharmacological activities as antimicrobial, antioxidant and anti-inflammatory (*Lončar et al., 2020*). The positive ionization based network using MS/MS data provided grouping of coumarins together in cluster B (Fig. S2). Coumarins are identified in LC-MS based on their characteristic loss of CO and $CO_2$ moieties (*Fu et al., 2020*). Herein, two coumarins were detected using positive ionization mode. Coumarin is a flavor compound in cassia cinnamon though of reported health hazards above a certain dose (*Farag et al., 2022*). Coumarin was detected in peak 69 with [M+H]⁺ at $m/z$ 147.0442 that yielded fragment ions at $m/z$ 103 [M+H-$CO_2$]⁺ and $m/z$ 91 [M+H-$CO_2$-CH]⁺ (*Farag et al., 2022*; *Fu et al., 2020*). Peak 73 showed [M+H]⁺ at $m/z$ 177.0548 (herniarin) (Fig. S4) with further fragmentation at $m/z$ 162 [M+H-$CH_3$]⁺, 133 [M+H-$CO_2$]⁺ and 121 [M+H-2CO]⁺ (*Fu et al., 2020*). Herniarin and coumarin were found to be more prominent in white mahlab than red seed cvs. It should be noted that herniarin was detected at relatively higher level than coumarin in white seed cvs. Both compounds were previously reported in most parts of *P. mahaleb*, viz. leaves, fruits, seeds and wood (*Ieri, Pinelli & Romani, 2012*). Coumarins are well-known to exhibit a variety of biological activities such as antioxidant, anticancer, antimicrobial actions (*Fylaktakidou et al., 2004*), although coumarins are nevertheless reported to cause hepatic damage and other negative impacts on health (*Farag et al., 2021*). Consequently, the quantity of coumarin in *P. mahaleb* should be assessed in future work to ensure its safety, especially for being consumed as food.

## Identification of cyanogenic glycosides

Cyanogenic glycosides are secondary metabolites that are widely spread in plant families such as Rosaceae (*Bolarinwa, Orfila & Morgan, 2014*) to play a defensive role upon insect attack aside from their health effects in humans. Two compounds were detected in negative versus one compound in positive ionization mode belonging to cyanogenic glycosides. Peak 44 with $[M-H]^-$ at *m/z* 456.1505 was identified as amygdalin based on fragment ion at *m/z* 323 resulting from the neutral loss of the disaccharide $[[M-H]^- -133]^-$ (*Otify et al., 2015*; *Xu et al., 2017*), whereas fragments appearing at 263 and 221 corresponding to cross-ring bond cleavage of glucose. Peak 42 with $[M-H]^-$ at *m/z* 502.1559 was identified as methoxy hydroxyl amygdalin (Fig. S5) with mass difference of 46 amu compared to peak 44 owing to the loss of methoxy group (30 amu) and hydroxyl group (16 amu) (*Otify et al., 2015*). Besides, an amygdalin derivative was also identified in positive mode ionization as prupersin E $[M+H]^+$ in peak 63 at *m/z* 563.1878 and in agreement with data in literature (*Chen et al., 2013*). Amgdalin and prupersin E were only identified in red mahlab and its roasted product, while methoxy hydroxyl amygdalin was detected in all seed accessions. Whether amgdalin and prupersin could serve as taxonomic markers for red seed mahlab has yet to be determined by analysing seeds from other resources. It should be noted that both methoxy hydroxy amygdalin and prupersin E were detected for the first time in mahlab cvs.

## Identification of flavonoids

Flavonoids are widely distributed in plants with health promoting activities to exist in plants as *O*-glycosides or *C*-glycosides. *O*-Glycosides are formed by attaching sugar to hydroxyl oxygen. While *C*-glycosides are sugar moieties combined directly to flavonoid backbone as C–C covalent bonds (*Xie et al., 2022*). Two flavonoid-*C*-glycosides first time to be reported in mahlab cvs. were detected in peaks 62 and 64 in red mahlab in negative and positive modes that showed characteristic fragmentation pattern in cross ring cleavages of the sugar moiety (−120 amu) or (−90 amu) compared to neutral losses of sugars in *O*-glycosides. The positive molecular network arranged both flavonoids in cluster E (Fig. S2). The ESI-MS spectra of peak 62 exhibited a molecular ion at *m/z* 431.0972 and fragmentation pattern of apigenin-*C*-hexoside (Fig. S6) with *m/z* 341 $[M-H-90]^-$, 311 $[M-H-120]^-$ (*Li et al., 2009*). Peak 64 exhibited molecular ion peak at *m/z* 461.1087 with fragmentation pattern for flavone methoxy-*C*-hexoside showing loss of 120 amu at *m/z* 371 and 90 amu at *m/z* 341 (*Shao et al., 2020*).

## Identification of amino acids and amino lipids

Positive ionization mode identified several free amino acids in all mahlab cvs. considering their improved detection in that mode due to nitrogen atom in amino acids. GNPS network aided in their identification (Fig. S2). Identified peaks included peak 1, 10, 11, 14, 15, 16, 20 with $[M+H]^+$ at *m/z* 175.1196 (arginine), 130.05 (pyroglutamic acid), 124.0393 (nicotinic acid), 268.1041 (adenosine), 182.0811 (tyrosine), 132.1019 (isoleucine) and 166.0865 (phenyl alanine), respectively. Arginine and nicotinic acid were only found in EGM and WEM. Pyroglutamic acid was detected in all mahlab cvs. and appeared to be

more abundant in EGM and WEM. Besides, white mahlab was previously examined for its amino acid profile and was found to exceed that of black mahlab with glutamic and aspartic acids as the major amino acids accounting for its nutritive value (*Mariod et al., 2009*). An amino acid derivative (peak 17) detected in WEM and REM with [M+H]$^+$ at *m/z* 276.1443 identified as glutaryl carnitine for the first time in mahlab seeds (Fig. S7) with fragmentation pattern as reported in (*Yang et al., 2020*). One acylated amino acid derivative (peak 19) was detected in red mahlab in negative ion mode, identified as deoxy fructosyl leucine with [M-H]$^-$ at *m/z* 294.1547. MS/MS Spectra showed subsequent loss of two water molecules (*Serag et al., 2020*). Other sugar acylated amino acid was detected in all mahlab seeds (peak 37) identified as tyrosinyl-*O*-hexoside with [M+H]$^+$ at *m/z* 344.1215 (*Abu-Reidah et al., 2015*). Tyrosinyl-*O*-hexoside was more prominent in EGM and WEM than REM and RREM. Deoxy fructosyl leucine and tyrosinyl-*O*-hexoside were identified for the first time in mahlab seeds. In addition, sugar acylated amino acids were the first time to be detected in genus *Prunus*.

Nitrogenous compounds were also detected in nonpolar peaks eluting later in chromatographic run as manifested by even weight i.e., peaks 87 and 94 with [M+H]$^+$ at *m/z* 284.3311 and 312.3626 identified as nonadecanamine and heptanamine, respectively. Nonadecanamine was detected in all mahlab cvs. in small quantity and reported for its antimicrobial activity (*Rivilla, 2021*), and whether it contributes to mahlab seed preservative effect in food has yet to be determined. Additionally, nonadecanamine and heptanamine were detected for the first time in mahlab cvs. and in genus *Prunus*.

## Identification of fatty acids and sphingolipids

Seeds are known to be rich in lipids as typical storage organs. Analysis of mahlab seeds revealed the presence of several fatty acids and sphingolipids in positive ionization mode appearing at Rt >10.00 min of UPLC chromatogram considering their lypophilic nature. Peak 81 with [M-H]$^-$ at *m/z* 293.2117 matched the fragmentation pattern of oxo-octadecatrienoic acid (*Xia et al., 2021*); a polyunsaturated fatty acid detected in white mahlab cultivars (EGM and WEM). Another fatty acid in peak 96 with [M+H]$^+$ at *m/z* 277.2164 identified as stearidonic acid (*Fernández-Ochoa et al., 2020*) was present in EGM and WEM. Peak 99 with [M-H]$^-$ at *m/z* 295.2268 showed fragmentation pattern similar to hydroxy linoleic acid (*Farag et al., 2022*). In addition, peak 105 was identified as octadecenedioic acid with [M-H]$^-$ at *m/z* 311.2229 (*Farag et al., 2022*). It's worth mentioning that those two fatty acids in peaks 99 and 105 were only detected in WEM. Also, it should be noted that all these fatty acids were first time to be identified in *P. mahaleb,* as previous work on different varieties of *P. mahaleb* resulted in the identification of omega-3 and 6 polyunsaturated fatty acids along with oleic acid. Accordingly, *P. mahaleb* is considered to exhibit a wide range of biological activities such as anti-inflammatory, antioxidant and to reduce the risk of atherosclerosis (*Farag et al., 2021*).

Another subclass of fatty acids present in mahlab seeds reported for the first time belonged to fatty acid ethanolamides, a family of endogenous lipid mediators with chemical structure consisting of fatty acid moiety linked to ethanolamine by amide linkage (*Angelini et al., 2017*). Fatty acid ethanolamides play key role in biological functions as analgesic,

anti-anxiety and promoting fat hydrolysis and weight loss (*Li et al., 2022*). The positive molecular network revealed the grouping of fatty acid ethanolamides in cluster C (Fig. S2).

Peak 101 with $[M+H]^+$ at *m/z* 324.2898 appeared in both EGM and WEM identified as linaloyl ethanolamide (Fig. S8) from key fragment ion at *m/z* 62 corresponding to ethanolamine moiety (*Li et al., 2022*) in addition to peaks 102 and 103. Linaloyl ethanolamide was reported to regulate sugar level and reduce pain (*Li et al., 2022*). Peak 102 with $[M+H]^+$ at *m/z* 300.2898 was identified as palmitoyl ethanolamide with fragment ion at *m/z* 282 owing to water loss and a fragment ion at *m/z* 62 for ethanolamine (*Angelini et al., 2017*). Likewise, peak 103 identified as oleoyl ethanolamide with $[M+H]^+$ at *m/z* 326.3055 with main fragment ion at *m/z* 62 showed similar fragmentation pattern (*Li et al., 2022*). Both fatty acid ethanolamides appeared only in EGM and found absent in WEM, REM and RREM. Fatty acid ethanolamides are the first time to be detected in genus *Prunus*. Oleoyl ethanolamide was reported to inhibit food intake, induce fat hydrolysis and reduce body weight (*Li et al., 2022*), and whether it contributes to an anti-obesity effect in *P. mahaleb* has yet to be determined.

Nitrogen containing lipids as sphingolipids and fatty acyl amides were found abundant in all mahlab cvs. except for RREM. Peaks 91, 93, and 95 were detected mostly in EGM and REM with an even mass weight in negative ionization mode with $[M+H]^+$ at *m/z* 564.3296, 522.356 and 566.3453 assigned as sphingolipids with a loss 285 amu indicative for sphingosine moiety (*Serag et al., 2020*) and aiding in the identification of that lipid subclass in mahlab seeds. Interestingly, fatty acyl amides are reported herein for the first time in mahlab seeds but not present in roasted mahlab likely attributed for their degradation at high temperature of roasting. Peak 78 showed an even mass weight of nonpolar fatty acyl amide identified as nonamide with $[M+H]^+$ at *m/z* 158.1543 (*Serag et al., 2020*). Peak 79 with $[M+H]^+$ at *m/z* 172.1695 was identified as decanamide (Fig. S9) with fragment ions at *m/z* 116 $[M+H-C_4H_8]^+$, 74 $[M+H-C_7H_{14}]^+$ (*Feng et al., 2020*). Peak 110 with $[M+H]^+$ at *m/z* 338.3413 was identified as docosenamide showing fragment ion at *m/z* 321 $[M+H-17]^+$, owing to the loss of ammonia (*El-Hawary et al., 2022*) and suggestive that mahlab seeds present potential source of shingolipids. Difference in fatty acyl amides level was detected *i.e.*, nonamide found more abundant than docosenamide in EGM, WEM and REM, decanamide was found exclusively in REM. In contrast, peaks 78, 79, 110 disappeared in RREM, probably due to roasting. Moreover, fatty acyl amides were identified for the first time in genus *Prunus*.

## Identification of arylalkylamide

Three arylalkylamides were detected for the first time in all mahlab seed cvs. at relatively high levels only in positive ion mode and highlighting the advantage of measuring in different ion modes. Peak 84, 89 and 97 with $[M-H]^-$ at *m/z* 304.3001, 332.3314 and 360.2636 were identified as benzyl tetradecanamine (Fig. S10), benzyl hexadecanamine and benzyl octadecylamine, respectively first time in mahlab seeds, and in agreement with reported spectra in literature (*Martin-Rivilla et al., 2020*). Interestingly, arylalkylamines were the first time to be reported in genus *Prunus*.

## Identification of miscellaneous compounds

Other metabolites that belong to different classes were identified for the first time in *P. mahaleb* seeds. Peak 2 was identified as choline at $[M+H]^+$ 104.1075 based on its main fragment ion at *m/z* 60. Choline; a potential functional food ingredient was identified as a major component in all seeds (*Holm et al., 2003*). Peak 6 was identified as hexosyl pentahydroxy heptanoic acid (Fig. S11) with $[M-H]^-$ at *m/z* 387.1144 showing fragments at *m/z* 341and 179 due to the successive loss of HCOOH (formic acid), $H_2O$, $C_4H_6O_4$ (succinic acid), hexosyl ($C_6H_{11}O_5$) and $CH_2O$ (*Nadeem et al., 2020*). Peak 7 at *m/z* 439.0857 detected in red mahlab was identified as dihydroxy phenol-*O*-galloyl-hexoside (*Zhu et al., 2016*). Norharman is $\beta$-carboline alkaloid that could be produced from roasting of protein rich food. Herein, it was identified in peak 46 with $[M+H]^+$ with *m/z* 169.076 only in roasted red mahlab (*Liu et al., 2022*) and likely to be considered a roasting marker.

Another phenolic aldehyde detected in seed and likely to contribute for seed flavor is vanillin detected in peak 49 in EGM and WEM with $[M+H]^+$ at *m/z* 153.0548, and almost absent in REM (*Shen et al., 2014*) suggestive that this platform provide insight on both mahlab aroma and secondary metabolites profile accounting for its food and health attributes.

## Multivariate data analysis of UPLC-ESI-MS dataset in negative and positive ionization modes

The UPLC-MS chromatograms revealed for qualitative and quantitative differences in metabolites classes among seed accessions, though a holistic assessment of metabolites was performed for further analysis of dataset using principal component analysis (PCA) in an untargeted manner. PCA is an unsupervised multivariate data analysis model that requires no classification of dataset that explained metabolite difference and aided in the unsupervised discrimination between seed accessions (*Farag et al., 2022*). The PCA model in negative ionization mode accounted for 71%of the total variance as prescribed by PC1 and PC2. The score plot (Fig. 2) showed clustering of white mahlab cvs. (EGM obtained from Greece and WEM obtained from Egypt) in one quadrant, while red mahlab (REM) alongside its roasted product (RREM) were at another quadrant, and suggestive that cv. type overcomes geographical origin in mahlab seeds. The loading plot (Fig. 2) revealed that phenolic acids, *i.e.*, coumaric acid hexoside, ferulic acid hexoside, ferulic acid hexoside dimer and dihydrocoumaroyl hexoside dimer accounted the most for seeds' segregation being enriched in white mahlab cv. from different locations. In contrast, red mahlab and its roasted product were clustered together being enriched in organic acids (citric and malic acid) and amygdalin derivative (methoxy hydroxyl amygdalin).

The dataset from positive ionization mode for the studied species was also subjected to PCA analysis to compare with that observed in negative ionization mode, with PC1 and PC2 to explain 66%of the variance and comparable to that in negative mode at 71%. Likewise, the score plot (Fig. 2) revealed that white mahlab cvs. (EGM and WEM) were segregated in one quadrant, while REM and RREM were clustered separately opposite side. The observed segregation was assigned to the enrichment of the two white mahlab cvs. in coumarins (coumarin and herniarin), hydroxycinnamate (ferulic acid) and nitrogenous compounds

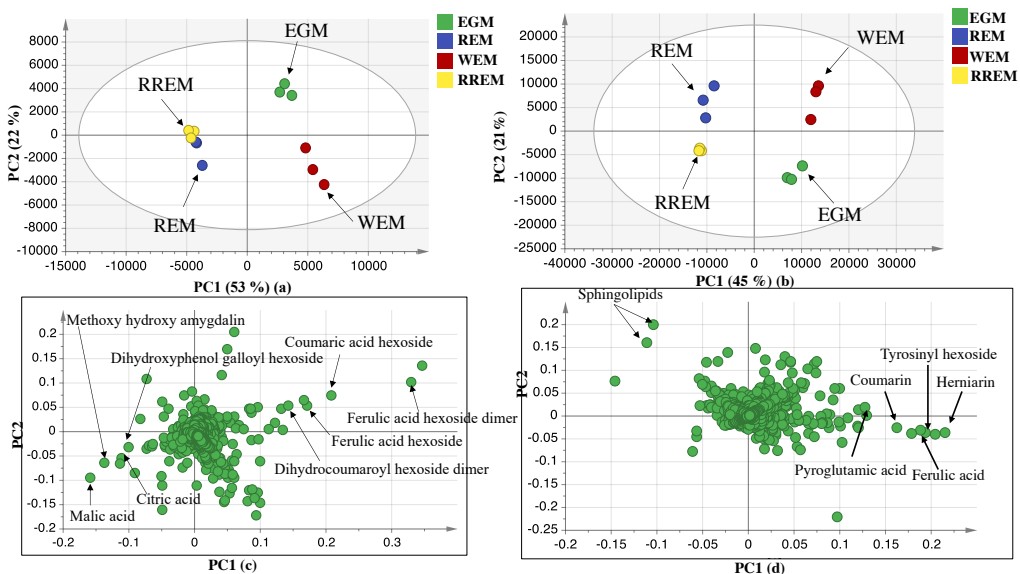

**Figure 2** **UPLC principal component analyses of the different cvs. of *Prunus mahaleb* L. seeds.** Negative ionization mode: (A) Score plot of PC1 vs. PC2, (B) respective loading plot with contributing mass peaks; positive ionization mode: (C) Score plot of PC1 vs. PC2, (D) respective loading plot with contributing mass peaks.

*viz.* pyroglutamic acid and amino sugar tyrosinyl hexoside. The main compounds that were responsible for the segregation of red mahlab and its roasted product are sphingolipids.

Supervised partial least squares-discriminant (OPLS-DA) analysis was further performed to ensure whether variant markers revealed using PCA could be used as potential markers for the examined cultivars. Data of red mahlab and its roasted product from negative and positive modes ionization were subjected to OPLS-DA analysis to identify markers that showed segregation of red cv. from others. OPLS-DA score plot for negative mode displayed model parameters R2 (goodness of fit) and Q2 (goodness of prediction) at 0.98 and 0.9, respectively. This model (Fig. 3) displayed that REM was more rich in ferulic acid-*O*-hexoside dimer. In contrast, OPLS-DA score plot for positive mode with R2 and Q2 at 0.97 and 0.94, respectively revealed that nitrogenous compounds accounted for the separation of REM from other seeds. The score plot (Fig. 3) showed that REM was enriched in nonamide, deoxy fructosyl leucine, glutaryl carnitine and isoleucine, while its roasted product (RREM) was more rich in choline.

## CONCLUSIONS

This study provided the first complete map of *Prunus mahaleb* L. cvs. metabolome accounting for its phytonutrient, primary metabolites and flavor compounds in that genus and to better account for its use as functional food. UPLC-MS led to the annotation of 110 compounds in both negative and positive ionization modes, of which most of the compounds (39) are first time to be detected in *Prunus mahaleb* L. White mahlab obtained from Egypt and Greece showed comparable metabolite pattern compared to red cv. as more
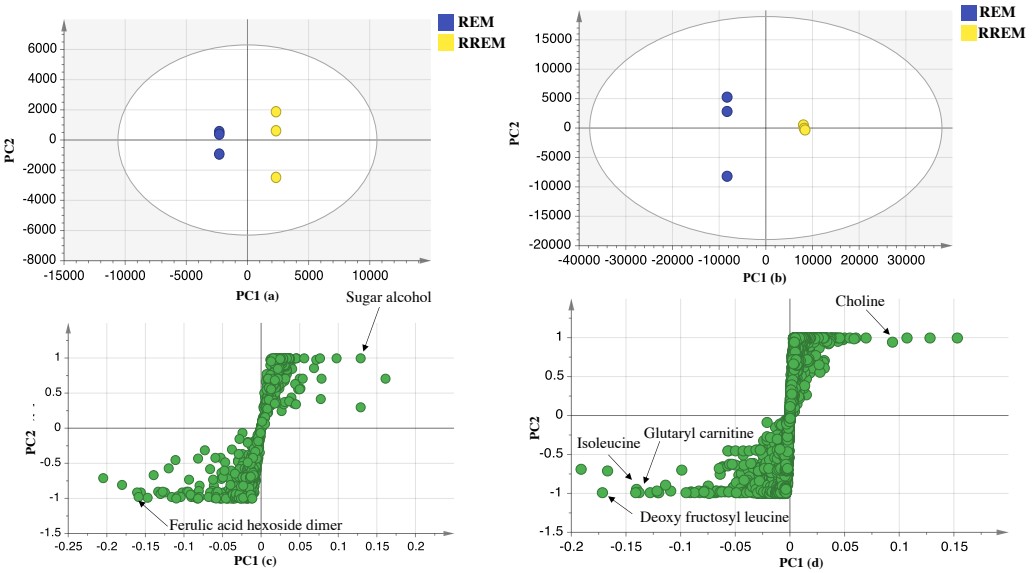

**Figure 3** **OPLS-DA score plot of REM and RREM.** In negative ion mode (A), in positive ion mode (B), loading S plot of REM and RREM in negative ion mode (C), in positive ion mode (D).

different. White cv. seeds were more rich in phenolic acids, with ferulic acid-*O*-hexoside dimer as the most abundant. In contrast, red cv. seed was more abundant in organic acids, with citric acid as a major component to account for seeds taste. Mahlab seeds metabolite profiling confirmed their valuable source of amino acids, polyunsaturated fatty acids and choline which adds to the nutritional value of mahlab. Multivariate analysis revealed that white mahlab from Greece and Egypt present a better source of coumarins to exhibit potential biological activities in addition to their characteristic taste and flavor. In contrast, red mahlab seeds appeared more enriched in sphingolipids alongside amino acids.

We do admit the limited selection of origins for *P. mahaleb* seeds; nevertheless, the same platform can be used for profiling seeds from other origin and to assess impact of agricultural practices, seasonal variation, growth stage and storage on seeds metabolome.

Isolation and identification of the metabolites should follow in the future for more conclusive work for their role in mahlab seeds potential as nutraceuticals. Moreover, more studies should be performed to investigate the biological activities of *Prunus mahaleb* L. in correlation with chemical components, and preferably from other resources for conclusive evidence for metabolites' heterogeneity in that genus.

### Funding
The authors received no funding for this work.

### Competing Interests
Mohamed A. Farag is an Academic Editor for PeerJ.

## Author Contributions

- Mayy M. Mostafa analyzed the data, prepared figures and/or tables, and approved the final draft.
- Mohamed A. Farag conceived and designed the experiments, performed the experiments, authored or reviewed drafts of the article, and approved the final draft.

## Data Availability

The raw data are available in the Supplemental File.

## Supplemental Information

Supplemental information for this article can be found online at http://dx.doi.org/10.7717/peerj.15908#supplemental-information.

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

# PeerJ

postharvest drying processes using UPLC-QTOF-MS. *ACS Omega* **6**:24484–24492 DOI 10.1021/acsomega.1c02926.

**Xie L, Deng Z, Zhang J, Dong H, Wang W, Xing B, Liu X. 2022.** Comparison of flavonoid O-glycoside, C-glycoside and their aglycones on antioxidant capacity and metabolism during *in vitro* digestion and *in vivo*. *Foods* **11**(**6**):882 DOI 10.3390/foods11060882.

**Xu S, Xu X, Yuan S, Liu H, Liu M, Zhang Y, Zhang H, Gao Y, Lin R, Li X. 2017.** Identification and analysis of amygdalin, neoamygdalin and amygdalin amide in different processed bitter almonds by HPLC-ESI-MS/MS and HPLC-DAD. *Molecules* **22**(**9**):1425 DOI 10.3390/molecules22091425.

**Yang L, Fang Y, Liu R, He J. 2020.** Phytochemical analysis, anti-inflammatory, and antioxidant activities of dendropanax dentiger roots. *BioMed Research International* **2020**:5084057 DOI 10.1155/2020/5084057.

**Yuan J, Wang Y, Mi S, Zhang J, Sun Y. 2021.** Rapid screening and characterization of caffeic acid metabolites in rats by UHPLC-Q-TOF mass spectrometry. *Tropical Journal of Pharmaceutical Research* **20**(**2**):389–401.

**Zhu T, Liu X, Wang X, Cao G, Qin K, Pei K, Zhu H, Cai H, Niu M, Cai B. 2016.** Profiling and analysis of multiple compounds in rhubarb decoction after processing by wine steaming using UHPLC—Q-TOF-MS coupled with multiple statistical strategies. *Journal of Separation Science* **39**:3081–3090 DOI 10.1002/jssc.201600256.