# Peer review of "Profiling of primary and phytonutrients in edible mahlab cherry (Prunus mahaleb L.) seeds in the context of its different cultivars and roasting as analyzed using molecular networking and chemometric tools"

_PeerJ, doi:10.7717/peerj.15908_

## Round 0.1 · original submission · Minor Revisions

Kindly revise your manuscript according to the reviewers' comments.
Make sure that all figures are of high quality; the discussion needs improvement, and previous studies and results need to be discussed in more details.

Reviewer 1 ·

Basic reporting

Prunus mahaleb L. (mahlab cherry) is a deciduous plant that is native to the Mediterranean region and central Europe with a myriad of medicinal, culinary and cosmetic uses. In this manuscript, the authors detected 110 primary and secondary metabolites from different cultivars of mahlab (white from Egypt and Greece, red from Egypt and post roasting), of which 39 are first time to be detected in Prunus mahaleb L, And assessed the variance of metabolites within seeds.The experiments are well conducted and presented. There are some special comments.The discussion section is less. It’s necessary for the author to have a thorough and comprehensive discussion.

Experimental design

Materials and Methods: More information about the plant material used is needed.
- How old were the trees that Prunus mahaleb L. seeds obtained from, All trees the same age?
- Where (in details) were the Prunus mahaleb L.seeds collected from?

Validity of the findings

No comment.

Additional comments

No comment.

·

Basic reporting

The article written can have better professional English

Experimental design

No Comment

Validity of the findings

No comment

Additional comments

No comment

Reviewer 3 ·

Basic reporting

1. Detailed scrutiny should be performed throughout the manuscript to revise/correct several grammar and stylistic issues.
2. Line 61-71: The justification, aim, and scope of this study are not clear in the Introduction section. In this regard, the authors must improve the reason for performing this phytochemical study and even its scope (e.g., further sample authentication).
3. Figure 1 and 2 should be improved in quality since the resolution is not good, and it is challenging to visualize the contained information.

Experimental design

1. The research question of this study is not clear. In addition, the manuscript does not state how the study fills an identified knowledge gap. The knowledge gap is not identified. It seems directed to the chemical study of P. mahaleb seeds from three cultivars, so the manuscript can go a little bit forward and provide a more robust aim and scope. In fact, lines 48-61 provide an idea about the chemical composition of P. mahaleb, but nothing is mentioned about varietal changes between cultivars or genotypes nor environmental factors-derived variations affecting the chemical composition. Therefore, I suggest rewriting the introduction and results and discussion sections to provide a good research question, a reasonable hypothesis, a good aim and scope, and a good discussion. As written, the aim and question of this study are not well-defined, relevant, and meaningful.
2. The number of samples is too low since two local cultivars (red and white), one external, and the roasted ones are insufficient to define adequate results. This can be rationalized due to a lack of an adequate aim and scope of this study. An appropriate scope for this low number of samples must be provided for readers.
3. In addition, the status (i.e., age, agronomic regime, fertilization, phytosanitary conditions, etc) of the parent plants of the harvested seeds of local origin is missing. In addition, the status of the external seeds must also be provided.
4. Revise in detail the Materials and Methods section. Some experimental details are missing to ensure outcome reproducibility. For instance: 1) Line 84: the grinding conditions to get seed powder, 2) line 94: MS analyzer is not mentioned, 3) Line 97: the processing details to get the data matrix from UPLC-MS data, 4) the type of MS data used (i.e., peak areas or peak intensities), 5) The model details and parameters to perform PCA and OPLS-DA analysis in SIMCA-P, 6) line 95: the manner to concatenate the positive and negative MS data, 7) line 106: the details and parameters used in Cytoscape to build the network are missing. All this information must be informed to readers. Detailed scrutiny must be performed to include relevant information to ensure outcome reproducibility for readers.
5. Quantifications related to umbelliferone (10 µg/mL) as an internal standard are not mentioned or exploited in this manuscript. Revise if this information is relevant to this study. Otherwise, It can be deleted.
6. Detailed scrutiny of the accurate mass values in the R&D section and Table 1 should be performed, looking for misleading and discrepancies between text and table.
7. Line 387: R2 and Q2, with 2 as superscript. Be consistent throughout the manuscript.
8. The discussion is highly descriptive and seems to be an extension of the results section, so the discussion can be improved since no references or comparisons to previous reports are provided.

Validity of the findings

1. Table 1: Revise the information between m/z values and the formula since they are not consistent. For instance, the m/z values seem to involve the quasimolecular ion (i.e., [M+H]+ and [M-H]-), but the formula seems to involve the molecular ion (i.e., M), so the charge is not justified in these formulas.
2. The discussion is highly descriptive and seems to be an extension of the results section, so the discussion should be improved since no references or comparisons to previous reports are provided.
3. The conclusion section should be improved since it summarizes the results. Authors should conclude the conceptual findings from a mechanistic point of view and even the scope of these results for future studies.

Additional comments

The manuscript in reference describes the phytochemical study of the seeds from several cultivars of Prunus mahaleb L. (mahlab) using UHPLC-MS/MS and GNPS platforms. The manuscript has relevant information, results, and organization that will interest readers. However, some points need to be addressed before further consideration.

---

## Round 0.2 · accepted · Accept

Dear Prof. Farag

Thanks for all the efforts done in revising your submitted manuscript and addressing all the reviewers' comments.

I am pleased with the current version which is completely suitable and ready for publication. Good Luck with all your future work.

Best Regards.